# Drugs That Induce Gingival Overgrowth Drive the Pro-Inflammatory Polarization of Macrophages In Vitro

**DOI:** 10.3390/ijms252111441

**Published:** 2024-10-24

**Authors:** Annalisa Palmieri, Agnese Pellati, Dorina Lauritano, Alberta Lucchese, Francesco Carinci, Luca Scapoli, Marcella Martinelli

**Affiliations:** 1Department of Medical and Surgical Sciences, University of Bologna, 40138 Bologna, Italy; annalisa.palmieri@unibo.it (A.P.); luca.scapoli2@unibo.it (L.S.); marcella.martinelli@unibo.it (M.M.); 2Department of Translational Medicine and for Romagna, University of Ferrara, 44121 Ferrara, Italy; agnese.pellati@unife.it (A.P.); dorina.lauritano@unife.it (D.L.); crc@unife.it (F.C.); 3Multidisciplinary Department of Medical-Surgical and Dental Specialties, University of Campania “Luigi Vanvitelli”, 80138 Naples, Italy

**Keywords:** DIGO, macrophage polarization, inflammation, gingival hyperplasia

## Abstract

Several attempts have been made to elucidate the pathogenesis of drug-induced gingival overgrowth (DIGO), which is triggered by the chronic use of certain drugs that fall into three main categories: anticonvulsants, immunosuppressants, and calcium channel blockers. Previous research suggests that cytokines and impaired cellular functions play a role in DIGO. Of particular interest are macrophages, immune cells that can switch between M1 (pro-inflammatory) and M2 (anti-inflammatory) phenotypes in response to exogenous signals and stimuli. An imbalance between M1 and M2 macrophage populations may underlie DIGO. M1 may contribute to the initial tissue damage in DIGO, while M2 may then attempt to repair the damage with anti-inflammatory mechanisms. To test the hypothesis that drugs associated with DIGO could influence macrophage polarization, human monocytes (precursors of macrophages) were induced to differentiate into M0-naïve macrophages and then exposed to drugs: diphenylhydantoin, gabapentin, mycophenolate, and amlodipine. Quantitative real-time PCR amplification was used to measure the expression of specific genes associated with macrophage polarization. All of the drugs tested induced M0 macrophages to overexpress genes typical of the M1 phenotype, such as CCL5, CXCL10, and IDO1. This investigation provides the first evidence of a link between drugs that cause DIGO and M1 pro-inflammatory macrophage polarization. The knowledge gained from this research could be valuable for future DIGO treatment strategies.

## 1. Introduction

Since 1939, it has been known that the chronic administration of phenytoin, an antiepileptic drug, can cause gingival overgrowth [1]. To date, many drugs administered for non-dental uses and falling into three different categories (antiepileptics, immunosuppressants and calcium channel inhibitors) have been implicated in this pathological condition, termed drug-induced gingival overgrowth (DIGO). It is characterized by excessive production in terms of the extracellular matrix, cellular hyperplasia, and/or hypertrophy in the gingival tissues. The incidence and prevalence of DIGO varies between drugs, as does the median time of onset [2].

DIGO has been explained as an alteration in the intracellular balance of sodium and calcium (the principle behind how these drugs function), leading to folic acid deficiency and the subsequent dysregulation of various cellular functions, resulting in an overgrowth. Extracellular matrix homeostasis is a delicate process involving several factors. Wang and colleagues proposed that one of these regulatory molecules could be Transient Receptor Potential Vanilloid-type 4 (TRPV4), a Ca^2+^-permeable plasma membrane channel that regulates collagen remodeling and which is involved in cell differentiation and proliferation, including fibrosis [3].

Droździk and Droździk proposed a major role for cytokines in mediating the response of innate and acquired immune systems to drugs, probably leading to epithelial–mesenchymal transition (EMT) [2]. Furthermore, the observed EMT of the gingival lining epithelium has been proposed to be responsible for the increased stromal consistency and increased extracellular matrix deposition [4], particularly in combination with calcium channel blockers [5]. However, it has also been suggested that DIGO may be caused by a reduction in collagen degradation rather than an increase in its production [6]. This is based on the observation that, in vitro, DIGO-derived fibroblasts decreased the expression of collagen-encoding mRNAs, resulting in reduced collagen and fibrosis [7,8]. This may be due to the reduction of intracellular folic acid, as reported above, which alters metalloprotease metabolism and prevents collagenase activation [9]. The pathophysiology of DIGO is complex, with many potential interacting factors leading to impaired cellular function [10]. As a result, it is difficult for researchers to explain all the details of the processes involved.

Cytokines appear to be responsible for a cascade of events leading to gingival overgrowth. Cytotypes that produce proliferative cytokines include fibroblasts, mast cells, and macrophages [11]. Beginning in embryonic development, macrophages are central players in tissue remodeling, homeostasis, and the restoration of tissue integrity after injury [12]. They can adopt different phenotypic states in response to endogenous or exogenous microenvironmental signals that trigger their polarization from naïve macrophages (M0) into the classically activated macrophage (M1), and the alternatively activated phenotype (M2). Mills proposed that macrophage plasticity is also expressed by the ability of M2 types to switch to M1 types, although M1 usually die or are replaced by monocyte-derived macrophages [13]. The M1 pro-inflammatory macrophages are activated by interferon-gamma (IFNγ) and lipopolysaccharide (LPS) and drive adaptive immune responses through the production of inflammatory cytokines such as TNF-α and IL-6. The M2 are tissue-resident macrophages responsible for tissue repair, cell proliferation, and ECM remodeling. When induced by interleukin (IL)-4/IL-13, they can produce large amounts of profibrotic cytokines, IL-10, TGF-β1, other growth factors, such as the platelet-derived growth factor (PDGF), vascular endothelial growth factor (VEGF), and FGF, and proteases including serine proteases, matrix metallopeptidase (MMP)-2, and MMP-9, which promote angiogenesis and ECM remodeling with anti-inflammatory effects [14,15].

A pivotal role in DIGO was attributed to specific macrophage phenotypes expressing platelet-derived growth factor-beta (PDGF-B) by Iacopino and colleagues in 1997 [16], but this was later disputed by Wright’s group [17].

Several researchers have hypothesized that the persistence of pro-inflammatory M1 macrophages and their reactive products may cause tissue damage that is compensated for through M2 polarization aimed at restoring the pre-inflammatory state and tissue healing, possibly with fibrosis formation [18,19].

Macrophage polarization has been considered relevant for several inflammatory, autoimmune, and allergic diseases [20]. This study was designed to test whether an imbalance of pro-inflammatory M1 macrophages and anti-inflammatory M2 macrophages could be involved in DIGO. Specifically, the experiments explored the ability of specific drugs, known to produce gingival hyperplasia, to induce the polarization of M0 macrophages. To the best of our knowledge, this type of investigation has not yet been described.

The drugs were selected on the basis of their therapeutic actions: diphenylhydantoin and gabapentin (anticonvulsants); mycophenolate (immunosuppressant); and amlodipine (calcium channel blocker) [21,22].

## 2. Results

### 2.1. Macrophage Polarization Assay

Monocytes were isolated from human peripheral blood using the density gradient separation method with Ficoll-Plaque and subsequently cultured for five days in Macrophage SFM culture medium containing the differentiation factor M-CSF to allow their differentiation into M0-naïve macrophages. The polarization of M0 macrophages into M1 and M2 macrophages was achieved after 48 h of treatment, with Macrophage SFM medium supplemented with the differentiation factors: lipopolysaccharide (LPS) and interferon-gamma (IFNγ) for the M1 phenotype; interleukin (IL)-4 for the M2 phenotype.

To investigate macrophage polarization after treatment with LPS + INFγ or IL-4, immunofluorescence was performed using a specific marker for the M1 or M2 phenotype. As shown in Figure 1, macrophages treated with LPS + INFγ (Figure 1c,d) were positive for CD80, a specific marker of the M1 subtype (Figure 1c). Macrophages treated with IL-4 (Figure 1e,f) were positive for CD163, which is specific for the M2 phenotype (Figure 1e).

The differentiation of M0 macrophages into the two subtypes, M1 and M2, was further assessed by monitoring the expression level of specific markers of polarization [23,24]. The expression profile of the selected markers, monitored by real-time RT-PCR, was able to discriminate between M1 and M2 polarization. As expected, IDO1, CXCL10, and CCL5 were significantly overexpressed in macrophages treated with LPS and IFNγ, revealing the differentiation of M0 macrophages into the M1 subtype (Figure 2a), while the overexpression of CD23, MRC1, and CCL22 characterized M2 differentiation driven by IL-4 (Figure 2b). The expression level fold changes and their statistical scores are shown in Appendix A.

### 2.2. Effect of Drugs on Macrophage Polarization

A preliminary test was performed to find the maximum concentration of the DIGO-inducing drugs that did not affect the viability of cultured macrophages. The following treatment concentrations were chosen: 100 µM for gabapentin, 10 µM for mycophenolate and diphenylhydantoin, and 1 µM for amlodipine.

The gene expression of the six markers that characterize macrophage polarization was measured by real-time PCR amplification. The significant overexpression of markers of M1 polarization, such as CCL5, CXCL10 and IDO1, was observed in all treatments. In contrast, none of the markers specific to the M2 phenotype (i.e., CD23, MRC1, or CCL22) showed relevant differences in expression levels (Figure 3a–d). Appendix A presents the expression level fold changes and statistical scores.

## 3. Discussion

Gingival enlargement is characterized by excessive periodontal tissue growth, due to a combination of gingival hyperplasia and hypertrophy. This condition is characterized by clinical symptoms such as pain, bleeding, abnormal tooth movement, periodontal disorders and aesthetic changes, but also occlusion problems, increased caries development, and periodontal disease.

Gingival overgrowth is one of the side effects caused by the chronic administration of therapeutic medications used for non-dental purposes, mostly belonging to the categories of anticonvulsants, immunosuppressants, and calcium channel blockers. In severe cases, the clinical crown of the dental elements is almost completely covered by gingival tissue. The severity of symptoms may depend on the duration and dose of the medication, the oral hygiene of the individual, as well as the degree of inflammation, fibrosis and cellularity [25].

Drug-induced gingival overgrowth (DIGO) and its chronic inflammation are maintained by the oral biofilm, which includes bacteria, viruses, and fungi that live in a homeostatic balance with each other and the immune system. Dysbiosis and an imbalance between oral bacteria and host pro-inflammatory and anti-inflammatory mediators are other factors that influence the severity of DIGO [26,27]. Other possible risk factors include poor plaque control, gender (with a three times higher risk for men than women), age (with an inverse correlation), and genetic predisposition.

The severity of DIGO seems to be influenced by plaque scores and gingival inflammation [21], and the 2014 classification system for periodontal diseases stated that plaque represents a cofactor in the etiology of DIGO [28]. Bacterial infections typically trigger pro-inflammatory cytokines in immune cells, which in turn stimulate fibroblasts to produce matrix metalloproteinases (MMPs) [29]. Moreover, the gingival overgrowth could make plaque control difficult, leading to a secondary inflammatory process, which aggravates the overgrowth induced by the drug [30].

Several lines of evidence suggest that macrophages may play a key role in DIGO by modulating the production of proliferative cytokines and the extracellular matrix with fibroblasts [2]. Naïve macrophages (M0) differentiate from circulating monocytes that have migrated into the connective tissue in response to chemotactic stimuli. M0 macrophages have a high degree of plasticity and can switch from one phenotype to another ex vivo by reprogramming that is mediated by inducer factors [12].

The aim of the present study was to investigate whether certain drugs belonging to the categories of anticonvulsants, immunosuppressants, and calcium channel blockers—which have gingival overgrowth as a side effect—are indeed able to induce the differentiation of macrophages into the pro-inflammatory M1 subtype. For this purpose, we isolated monocytes from human peripheral blood and induced them to differentiate into macrophages. The M0 macrophages were then treated for 48 h with diphenylhydantoin, gabapentin, mycophenolate or amlodipine—all of which are known to induce gingival overgrowth. The analysis of polarization markers revealed a consistent increase in the expression of genes typical of the M1 subtype. In fact, after 48 h of treatment, IDO1 was the most highly expressed gene for all four drugs tested, with a fold change increase ranging from 4.88 to 12.91. The other two genes typical of the M1 phenotype (CCL5 and CXCL10) were also significantly upregulated after each of the four treatments, confirming the reprogramming from M0 to M1 triggered by the selected drugs.

It is widely accepted that the polarization of macrophages into the M1 type, which is accompanied by the release of inflammatory cytokines (e.g., IL-6, IL-12, TNF-α), reactive oxygen species, and antimicrobial peptides, is an important indicator of inflammation [31]. This process is believed to be a crucial step in the transition to the wound-healing phase, which is supported by the involvement of M2-type macrophages.

Based on our findings, we postulate that the drug-induced activation of pro-inflammatory M1 macrophages may potentially lead to a chronic condition favored by the continued recruitment of M2 macrophages: these cells secrete TGF-β [19], a cytokine capable of stimulating fibroblast proliferation and excessive deposition of ECM, elements typically observed in gingival overgrowth.

Situations in which medical therapies cannot be modified represent a significant challenge in clinical practice, as gingival overgrowth often recurs even after periodontal surgical excision. Alternatively, the use of azithromycin has been suggested as a pharmacological strategy for its indirect anti-inflammatory role, mediated by its antimicrobial properties. However, this strategy requires high doses, and operates by an unclear mechanism [32]. A preliminary study in mice suggests that statins may have a role in the prevention or attenuation of phenytoin-induced human gingival growth by modulating TGF-β1-induced CCN2 expression and EMT [33]. In light of this, it is clear that the inhibition of M1 macrophage polarization could be a key strategy for the treatment of DIGO. A first attempt should be made with those naturally occurring compounds that have already been shown to be able to regulate M0/M1 polarization in various tissues. These include diosgenin glucoside, which suppresses the expression of M1 markers in microglial cells stimulated with LPS; osbeck and luteolin, which inhibit M1 polarization by suppressing the NF-κB pathway; and lupeol, which inhibits M1 polarization by downregulating the expression of IRF5, a key transcription factor in M1 polarization [34].

The study described here is a preliminary study with some limitations. It explored only the very first step in the pathogenesis of DIGO, i.e., the polarization of M0 to M1 macrophages, while the cascade effect of M1 polarization on M2 induction and stromal component remodeling was not evaluated. In addition, we could not account for the possible action of the bacterial flora in inducing an inflammatory state in synergy with drugs.

The validation of the hypothesized mechanisms leading to gingival overgrowth in an in vivo model is worthy of further investigation, with the goal of developing interventions that limit gingival enlargement, a condition that leads to the worsening of patients’ quality of life.

## 4. Materials and Methods

### 4.1. Isolation of Monocytes from Peripheral Blood, Differentiation into Macrophages and Their Polarization

Two of the manuscript authors who provided signed informed consent donated fresh whole blood for monocyte isolation. The donors were a 47-year-old woman and a 58-year-old man. Isolation was performed by density gradient centrifugation using Ficoll-Paque Premium 1.073 (GE Healthcare, Chicago, IL, USA). The number of isolated monocytes was approximately 500,000 cells/mL in the blood.

Monocytes were resuspended in Macrophage Serum Free Medium (Macrophage SFM) (Thermo-Fischer, Waltham, MA, USA) and supplemented with 100 ng/mL of Macrophage Colony-Stimulating Factor (M-CSF), a differentiation factor (Sigma-Aldrich, St. Louis, MO, USA), 1% penicillin/streptomycin (Sigma-Aldrich, St. Louis, MO, USA), and 1% l-glutamine (Sigma-Aldrich, St. Louis, MO, USA), then plated in 12-well cell culture plates. Each well contained 1,000,000 monocytes obtained from approximately 2 mL of whole blood.

Monocytes were cultured in an incubator at 37 °C and 5% CO_2_ for five days until they adhered to the substrate and could be considered M0 macrophages. After 5 days, M0 macrophages were treated for 48 h to induce the polarization into M1 and M2 macrophages. The treatments consisted of lipopolysaccharide (LPS) and interferon-gamma (IFNγ) (Sigma-Aldrich, St. Louis, MO, USA), both at a final concentration of 100 ng/mL for M1 polarization, and interleukin 4 (IL-4) (Sigma-Aldrich, St. Louis, MO, USA) at a final concentration of 40 ng/mL for M2 polarization.

### 4.2. Immunofluorescence

Adherent macrophages were fixed with 4% paraformaldehyde for 10 minutes at room temperature. The cells were then permeabilized with 0.1% Triton X-100 (Sigma-Aldrich, St. Louis, MO, USA), and non-specific reactive sites were blocked with 10% bovine serum albumin (BSA) solution (Sigma-Aldrich, St. Louis, MO, USA). Cells were then incubated overnight at 4 °C with mouse anti-human CD80 monoclonal antibody (Thermo Fischer Scientific, Waltham, MA, USA) and rabbit anti-human CD163 monoclonal antibody (Thermo Fischer Scientific, Waltham, MA, USA). Detection was performed using TRITC-conjugated goat anti-mouse CD80 (Sigma-Aldrich, St. Louis, MO, USA) and FITC-conjugated goat anti-rabbit CD163 (Sigma-Aldrich, St. Louis, MO, USA) (60 minutes at room temperature). Negative control experiments were performed by omitting primary antibodies. Finally, cells were mounted with DAPI (Vector Laboratories, Burlingame, CA, USA) and observed under a fluorescence microscope (Eclipse TE 2000-E, Nikon Instruments S.p.a., Florence, Italy).

### 4.3. Cell Viability Assay

Stock solutions for each drug—diphenylhydantoin, gabapentin, amlodipine and micophenolate (Merck, Darmstadt, Germany)—were prepared at a concentration of 10 mM by dissolving the substances in the specific solvent: water for diphenylhydantoin and gabapentin, DMSO for the others. Serial dilutions of each 10 mM stock solution were prepared using Macrophage SFM culture medium supplemented with antibiotics and amino acids, obtaining four solutions with the following concentrations: 100 µM, 10 µM, 1 µM, 0.1 µM. Cell viability assays were performed using PrestoBlue™ reagent (Invitrogen, Carlsbad, CA, USA). Macrophages were seeded in 96-well plates at a density of 5000 cells/well, and incubated for 24 h to allow cell attachment. Serial dilutions of each drug solution were added (three wells for each concentration). The cell culture medium alone was used as a negative control.

After 48 h of incubation, cell viability was measured using the PrestoBlue™ reagent (Invitrogen, Carlsbad, CA, USA) according to the manufacturer’s instructions. The absorbance was measured at 570 nm excitation and 620 nm emission wavelengths using an automated microplate reader (Sunrise™, Tecan Trading AG, Männedorf, Switzerland). The percentage of viable cells was determined by comparing the mean absorbance in drug-treated wells with the mean absorbance in control wells exposed to the vehicle alone. A concentration sufficient to study the biological effects of treatment without reducing cell viability below 80% was chosen for each drug (Figure 4).

### 4.4. Treatment of Naïve Macrophages with Drugs

After determining the most appropriate concentration for the treatments, the macrophages were seeded in 12-well cell culture plates and treated for 48 h with Macrophage SFM medium solution containing the drug to be tested at the following concentrations: gabapentin 100 µM; mycophenolate 10 µM; amlodipine 1 µM; diphenylhydantoin 10 µM. The control was Macrophage SFM medium containing the same amount of DMSO or water, which was used to dissolve the agents. After 48 h of exposure, RNA was extracted from the cells.

### 4.5. RNA Extraction and Reverse Transcription

Total RNA was extracted from macrophages using the RNeasy Mini Kit (Qiagen, Hilden, Germany), according to the manufacturer’s protocol, and quantified using a Nanodrop 2000 spectrophotometer (Thermo Scientific, Waltham, MA, USA). A 500 ng aliquot of RNA for each sample was reverse-transcribed to cDNA using the PrimeScript™ RT Master Mix kit (Takara Bio, Kusatsu, Shiga, Japan). The thermal profile of the reaction included a step at 37 °C for 15 min and reverse transcription inactivation at 85 °C for 5 s.

### 4.6. Real-Time PCR Amplification

To assess the polarization of macrophages toward M1 and M2 subtypes, the expression level of specific markers was assessed by real-time PCR. Markers of M1 polarization were CXCL10 (C-X-C motif chemokine ligand 10), CCL5 (C-C Motif Chemokine Ligand 5), and IDO1 (Indoleamine 2,3-Dioxygenase 1), while markers of M2 polarization were CCL22 (C-C Motif Chemokine Ligand 22), MRC1 (Mannose Receptor C-Type 1) and CD23 (Fc epsilon receptor II) [23,24]. Primers and dual-labeled probes for each target were designed with the aid of primer-BLAST online tool: sequences are shown in Table 1 and Table 2.

Real-time PCR amplifications were performed in a final volume of 20 μL containing 10 μL of 2X TaqMan™ Gene Expression Master Mix (Applied Biosystems™, Foster City, CA, USA), 400 nM of each primer, 200 nM of probe, and 150 ng of cDNA. The reaction was performed on an ABI PRISM 7500 instrument (Applied Biosystems, Foster City, CA, USA). After an initial denaturation step at 95 °C for 2 min, amplification was performed for 40 cycles of a two-step profile of 15 s at 95 °C and 60 s at 60 °C.

Each reaction was carried out in analytical duplicate, and a negative control was used in each assay to exclude any biological contamination. Gene expression levels were normalized to those of the reference gene RACK1 (Ras-related C3 botulinum toxin substrate 1), labeled with cyanine 5 (Cy5) (forward sequence: 5′-CGGTGAATCTGGGCTTATGGGA-3′; reverse sequence: 5′-GGAGGTTATATCCTTACCGTACG-3′; probe sequence: 5′-TCCTCTCCGCCTCTCGAGATAAGACCA-3′).

### 4.7. Statistical Analysis

The relative gene expression was quantified with the delta/delta Ct calculation method [35], using the reference gene RACK1 to normalize gene expression levels. Paired sample t-test was employed to compare the delta Ct values between treated macrophages and the controls. The mean expression levels of M1 or M2 subtypes were calculated as fold changes compared to the expression levels of M0 macrophages. The gene’s expression level change in response to the treatment was considered biologically relevant when the expression level was doubled (fold changes ≥ 2, *p* value ≤ 0.05) or halved (fold changes ≤ 0.5, *p* value ≤ 0.05).

## 5. Conclusions

To the best of our knowledge, no evidence has previously been reported linking drugs that induce gingival overgrowth with pro-inflammatory M1 macrophage polarization. It has not yet been demonstrated how the overactivation of M1 can lead to gingival overgrowth. However, M1 polarization could in fact represent the necessary step to activate the inflammatory process, leading to the release of molecules such as cytokines (IL-4/IL-13), which in turn induce the polarization from M0 to M2 needed to restore the tissue environment using anti-inflammatory mechanisms [14].

The evidence reported in this study requires confirmation and verification of the process in vivo; however, it may serve as a useful starting point for understanding the mechanisms of certain drugs in inducing DIGO.

## Figures and Tables

**Figure 1 ijms-25-11441-f001:**
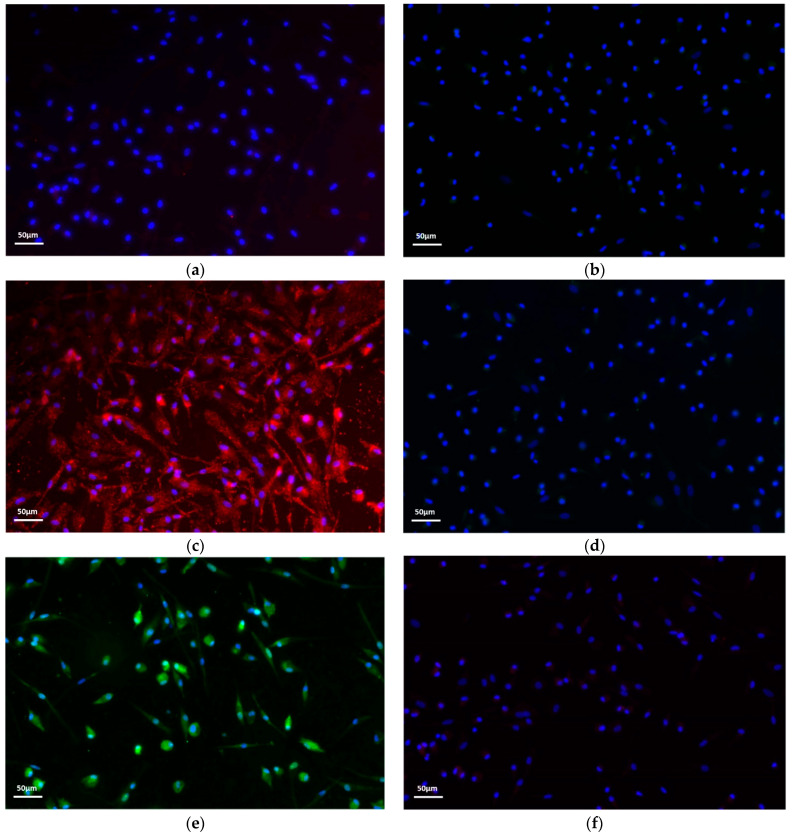
Immunofluorescence staining of the adhered macrophages after differentiation. Magnification 20X. (**a**,**b**) M0 macrophages negative for CD80 and CD163, respectively; (**c**) M1 macrophages positive for CD80 (red); (**d**) M1 macrophages negative for CD163; (**e**) M2 macrophages positive for CD163 (green); (**f**) M2 macrophages negative for CD80. Nuclei were stained with DAPI (blue).

**Figure 2 ijms-25-11441-f002:**
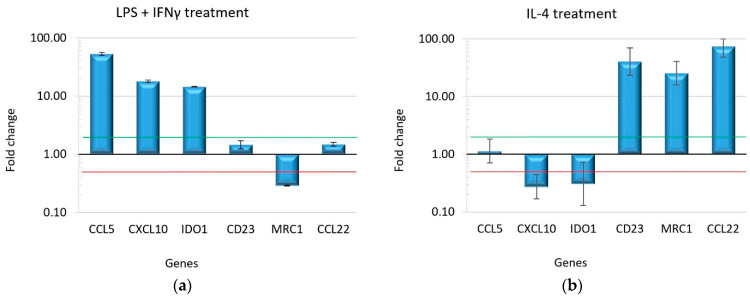
Gene expression profile in differentiated macrophage subtypes M1 and M2: (**a**) macrophages treated with LPS and IFNγ showed the overexpression of the M1 markers and downregulation of the M2 markers; (**b**) IL-4 treatment induces the overexpression of the M2 markers. The bars in the graph represent the fold change in gene expression on a logarithmic scale. The error bars indicate the standard deviation of fold changes calculated from two biological samples and three experimental replicates. The green line marks a fold change = 2; the red line marks a fold change = 0.5.

**Figure 3 ijms-25-11441-f003:**
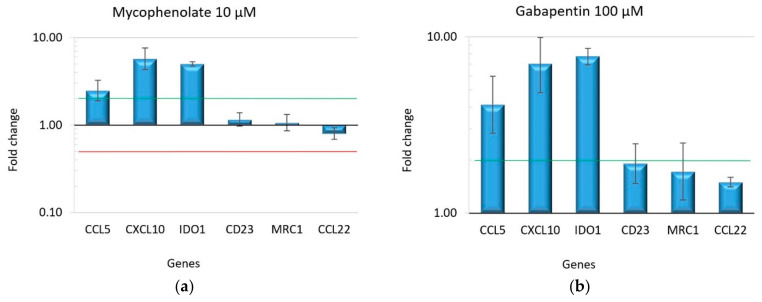
The gene expression profile of macrophages treated with different drugs after 48 h. Panel (**a**) shows the results of gabapentin treatment, panel (**b**) of mycophenolate, panel (**c**) of amlodipine, and panel (**d**) of diphenylhydantoin. All the treatments induced M1 polarization, as evidenced by the significant upregulation of CCL5, CXCL10, and IDO1 genes. The bars in the graph represent the fold change in gene expression on a logarithmic scale. The error bars indicate the standard deviation of fold changes calculated from two biological samples and three experimental replications. The green line marks a fold change = 2; the red line marks a fold change = 0.5.

**Figure 4 ijms-25-11441-f004:**
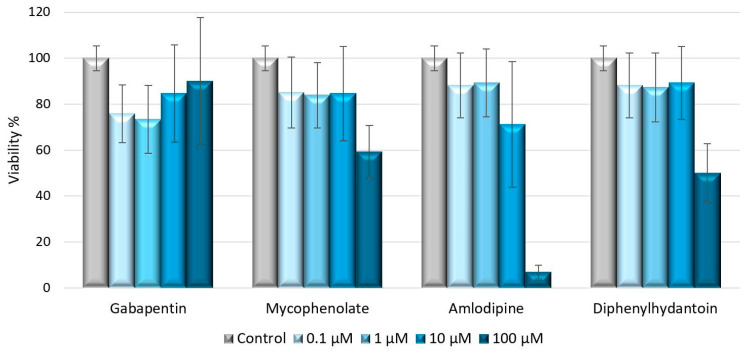
Cell viability of macrophages (M0) treated for 48 h with different concentrations of drugs, assessed by PrestoBlue™ reagent protocol. The viability of treated samples was normalized to untreated control; error bars represent standard errors calculated from three experimental replications.

**Table 1 ijms-25-11441-t001:** Sequences of the primers and probes of the markers chosen to characterize macrophages differentiated into the M1 subtype.

Gene	Primer Forward 5′-3′	Primer Reverse 5′-3′	Probe 5′-3′
IDO1	CAGCTTCGAGAAAGAGTTTGATGA	CTTTCACGTTCTTTGTTCTCAGGT	TGCCTGTGAGTCCGATTTCTGAGGCTGA—*FAM*
CXCL10	AGCCTCTGTGTGGTCCAT	TCGAAGGCCATCAAGAATTTA	ACTGCATCGATTTTGCTCCCCTCTGGT—*FAM*
CCL5	GCAAGTCTGGCAGGATTTCC	ACACACTTGGCGGTTCTTTC	TGACTCCCGGCTGAACAAGGGCAA—*FAM*

**Table 2 ijms-25-11441-t002:** Sequences of the primers and probes of the markers chosen to characterize macrophages differentiated into the M2 subtype.

Gene	Primer Forward 5′-3′	Primer Reverse 5′-3′	Probe 5′-3′
CCL22	TGGCGCTTCAAGCAACTG	TAGAAGTGTTTCACCACGCG	GCGTCTGCTGCCGTGATTACGTCC—*FAM*
MRC1	TCGAGGAAGAGGTTCGGTTC	CTCCAATCCCGGTTCTCATG	CCCACTGGAATTCAGTATGCCAGGGCG—*FAM*
CD23	TATGCCTGTGACGACATGGA	GCATGCGTCAGGAAGTC	CTGGTCAGCATCCACAGCCCGGAG—*FAM*

## Data Availability

The data that support the findings of this study are available from the corresponding author, (A.L.), upon reasonable request.

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
