# Peer review of "Drugs That Induce Gingival Overgrowth Drive the Pro-Inflammatory Polarization of Macrophages In Vitro"

_ijms, 2024, doi:10.3390/ijms252111441_

Round 1
Reviewer 1 Report
Comments and Suggestions for Authors
This manuscript, titled addresses a relevant topic in oral pathology, exploring the potential link between drugs known to induce gingival hyperplasia and macrophage polarization.
The authors have conducted a well-structured in vitro analysis, examining the effects of four drugs from different therapeutic classes on macrophage polarization. The methodology is described clearly and in detail, allowing for experimental reproducibility. The use of specific markers to identify M1 and M2 macrophage phenotypes strengthens the reliability of the results. The discussion effectively contextualizes the results within the current understanding of DIGO pathogenesis. The authors appropriately acknowledge the limitations of their in vitro study and suggest future directions for research, including the need for in vivo verification. While the study provides valuable insights, it would benefit from a more detailed exploration of the potential mechanisms by which M1 polarization might lead to gingival overgrowth. Discussion of the clinical implications of these findings, particularly in terms of potential therapeutic interventions. Consideration of how other factors, such as dental plaque or systemic conditions, might interact with the observed drug effects on macrophage polarization in vivo. Additionally, a limitation of the manuscript is its references. The current bibliography is both limited in number and relies heavily on older sources. To strengthen the paper's relevance and situate it within the most current understanding of the field, the authors should substantially update their references. This should include recent studies on macrophage polarization, DIGO pathogenesis, and related topics published within the last 5-7 years.
Overall, this study represents a significant contribution to the field, offering a new perspective on the pathogenesis of DIGO and potentially opening new avenues for therapeutic approaches. The manuscript is well-written, logically structured.
Comments on the Quality of English LanguageMinor editing is required
Reviewer 2 Report
Comments and Suggestions for Authors
Title
The title is clear and captures the study’s main focus.
Abstract
The abstract is clear, however the authors should add the key methodology used in this study. In addition, to give a stronger sense of the results they should specify the fold change of key genes.
Introduction
The authors should emphasize the role of M2 macrophages in tissue repair.
The hypothesis about drugs inducing M1 polarization is clear, however the authors should explicitly what knowledge gaps this study aims to fill.
Methods
The authors should make sure that the ethical approvement of a specific committee is not necessary. They should add in the letter of reviewer answer the reference law.
Results
The statistical significance is mentioned in terms of fold change, however I suggest using the usual annotation with * and p value.
Discussion
The discussion synthetize the results without integrating them with literature. I suggest revising this section.
The authors should add the limitations and future directions of this study.
Reviewer 3 Report
Comments and Suggestions for Authors
The present experimental study tested the hypothesis that drugs associated with DIGO could influence macrophage polarization. Human monocytes were differentiated into M0 naïve macrophages and then exposed to drugs: diphenylhydantoin, gabapentin, mycophenolate, and amlodipine. The authors found that all of the drugs tested induced a significant overexpression of genes characteristic of the M1 phenotype in M0 macrophages. They claimed that the present study is the first evidence of a link between drugs that cause DIGO and M1 pro-inflammatory macrophage polarization. In addition, that the knowledge gained from this research could be valuable for future treatment strategies of DIGO.
General aspects:
This is a well-written and interesting paper that examined the effect of DIGO-associated drug on MO polarization. This is a narrow scientific field studied rather intensively several decades ago. In these studies, the gingival fibroblasts, periodontal ligament and oral epithelial cells were the targets for the drug effects. In addition, the effects of microbial challenge in combination with the drugs has been a central dogma. The results from the present study clearly showed that the drugs targeted genes that promotes differentiation towards pro-inflammatory M1 MO. A limitation of the study as lack of data that indicate interaction with bacterial challenge and effect on periodontal cell proliferation. Monocytes that are recruited to the gingiva from the peripheral circulation meet a gradient of microbial and inflammatory compounds upon migration towards the inflamed gingiva. The overgrowth in the gingiva is restricted to the periodontal tissue cells. These aspects can be addressed by additional experiments or further discussions about these cellular and molecular processes.
Specific points:
Line 63-75 The selection of genes for Taqman is relevant for M! polarization, however, it has not given any information concerning pro-inflammatory cytokines. This link needs to be further explained in the revised introduction.
Line 82-93 How many different donors were involved, age and sex? Add information in the section material method line 156-166.
Line 94-97 Figure 1 legend. Is this a representative experiment, or the only that was performed?
Line 113-116 How are the selected concentrations of the drugs related to clinical use. This has to be addressed in the manuscript.
Line 122 Results from protein analyses of pro-inflammatory cytokines released in the culture supernatant should contribute significantly to the importance of the study. Methods for the selected genes for mRNA expression are available, both as ELISA or Western blot. A protein profiler for inflammation could also be an alternative.
Line 122-128 What are the spread measure based on, SD? How many experiments and repeats? Add information to the figure legend.
Line 130-152 Add limitations of the study also in the discussion. Not only in conclusion.
Line 208 Figure 4 needs revision. Put the lowest concentration next to the control, and the highest concentration to the right at the x-axis.
Line 210 How many replica or experiments are the SD based on? Add information to the figure legend.
Line 252-257 Was there any statistic calculation made for the cytotoxicity analyses? Information that could be added in a revised manuscript.
To conclude: I suggest that the authors confirm gene expression analyses of cytokine/chemokines in the culture supernatants before the manuscript can be published. It is not always that mRNA expression results in protein production and release. In addition, it is not described in the manuscript how many experiments that have been performed for each setting.
Round 2
Reviewer 2 Report
Comments and Suggestions for Authors
The manuscript can be accepted.
Reviewer 3 Report
Comments and Suggestions for Authors
The review report are fine and I have no further comments or suggestions to the revised manuscript.